# Accounting for heterogeneity due to environmental sources in meta-analysis of genome-wide association studies
Siru Wang[1,11] ✉, Oyesola O. Ojewunmi[2,11], Abram Kamiza[3,4,5], Michele Ramsay [3], Andrew P. Morris[6], Tinashe Chikowore[7,8,9,12], Segun Fatumo[2,4,10,12] & Jennifer L. Asimit [1,12] ✉

Meta-analysis of genome-wide association studies (GWAS) across diverse populations offers power gains to identify loci associated with complex traits and diseases. Often heterogeneity in effect sizes across populations will be correlated with genetic ancestry and environmental exposures (e.g. lifestyle factors). We present an environment-adjusted meta-regression model (env-MR-MEGA) to detect genetic associations by adjusting for and quantifying environmental and ancestral heterogeneity between populations. In simulations, env-MR-MEGA has similar or greater association power than MR-MEGA, with notable gains when the environmental factor has a greater correlation with the trait than ancestry. In our analysis of low-density lipoprotein cholesterol in ~19,000 individuals across twelve sex-stratified GWAS from Africa, adjusting for sex, BMI, and urban status, we identify additional heterogeneity beyond ancestral effects for seven variants. Env-MR-MEGA provides an approach to account for environmental effects using summary-level data, making it a useful tool for meta-analyses without the need to share individual-level data.

Genome-wide association studies (GWAS) have identified thousands of loci contributing genetic effects to a wide range of complex traits and diseases[1–4]. Although GWAS have successfully identified hundreds of genetic variants associated with complex diseases, such genetic variants explain a small proportion of heritability[5,6]; many variants with smaller effect sizes explain the remaining contribution. To identify novel variants with modest effects, meta-analysis is a common approach to collect GWAS and obtain large sample sizes without sharing individual-level genotype and phenotype data.

When conducting a meta-analysis, multiple GWAS are usually collected from genetically similar studies to minimise heterogeneity in the allelic effect at each genetic variant across these GWAS[7,8]. Under this assumption, some fixed-effects models were developed[9–12]. However, even when studies are from the same genetic ancestry, heterogeneity in the effect sizes inevitably occurs due to genetic ancestry and environmental exposures that may interact with genetic variants; the fixed-effect analysis cannot

accommodate this heterogeneity issue[13]. Some causes of allelic heterogeneity include variations in linkage disequilibrium (LD) patterns between cohorts and varied environmental exposures across cohorts that could interact with genetic contributions[14–18].

Another approach for addressing heterogeneous effect sizes is meta-analysis with a random-effects model[19–21]. Traditional random-effects models estimate the effect sizes and their standard errors by assuming that the effect sizes from the studies follow the same distribution[22]. However, when variants have varying effect sizes across cohorts, it has been demonstrated that traditional random-effects models have lower power to detect associations than fixed-effects models[23–25]. An alternative random-effects model[25] has been shown to have power gains over fixed-effects models by assuming that there is no heterogeneity under the null hypothesis. Nevertheless, these approaches do not take into account that the extent of heterogeneity between a pair of GWAS is likely to be correlated with the genetic

[1]MRC Biostatistics Unit, University of Cambridge, Cambridge, UK. [2]Department of Non-Communicable Disease Epidemiology, London School of Hygiene and Tropical Medicine, London, UK. [3]Sydney Brenner Institute for Molecular Bioscience, Faculty of Health Sciences, University of the Witwatersrand, Johannesburg, South Africa. [4]The African Computational Genomic (TACG) Research Group, MRC/UVRI and LSHTM, Entebbe, Uganda. [5]Malawi Epidemiology and Intervention Research Unit, Lilongwe, Malawi. [6]Centre for Genetics and Genomics Versus Arthritis, Centre for Musculoskeletal Research, The University of Manchester, Manchester, UK. [7]MRC/Wits Developmental Pathways for Health Research Unit, Department of Paediatrics, Faculty of Health Sciences, University of the Witwatersrand, Johannesburg, South Africa. [8]Channing Division of Network Medicine, Brigham and Women's Hospital, Boston, MA, USA. [9]Harvard Medical School, Boston, MA, USA. [10]Precision Healthcare University Research Institute, Queen Mary University of London, London, UK. [11]These authors contributed equally: Siru Wang, Oyesola O. Ojewunmi.[12]These authors jointly supervised this work: Tinashe Chikowore, Segun Fatumo, Jennifer L Asimit.
✉e-mail: siru.wang@mrc-bsu.cam.ac.uk; jennifer.asimit@mrc-bsu.cam.ac.uk

distance between them. That is, GWAS from genetically similar ancestry groups will be more homogenous than those from diverse ancestry groups.

To account for the heterogeneity between ancestry groups, MANTRA (Meta-ANalysis of Transethnic Association studies) uses a Bayesian framework that clusters different cohorts by a prior model of genetic similarity, which is assessed by mean pairwise genome-wide allele frequency differences[26]. Compared to the fixed-effects and random-effects meta-analysis, MANTRA has shown significant improvement in power to detect associations when the similarity in allelic effects between populations is well-captured by their relatedness[27]. However, as MANTRA implements a Metropolis–Hastings Markov chain Monte Carlo algorithm[28,29], it is not scalable for the meta-analysis of many GWAS. To address this computational challenge, MR-MEGA (Meta-Regression of Multi-Ethnic Genetic Association)[30] was developed, to account for and quantify the heterogeneity in allelic effects that is due to genetic ancestry, whilst testing for genetic associations. MR-MEGA models allelic effects, weighted by the corresponding standard errors, as a function of axes of genetic variation derived from mean pairwise genome-wide allele frequency differences between GWAS. Compared to fixed and random-effects meta-analysis, MR-MEGA has increased power to detect association when there is heterogeneity in allelic effects between populations due to genetic ancestry[30].

Increasing evidence shows that different environmental exposures between GWAS may contribute to varying allelic effects across populations. For example, sex can be treated as a simple "environmental" risk factor, incorporating physiological and behavioural differences between males and females at different stages, possibly leading to differences in allelic effects between sexes[31]. A series of examples of sex-differentiated effects were confirmed through human GWAS[32–34], demonstrating that sex can interact with causal variants, which causes heterogeneity in allelic effects between males and females.

Considering the impact of differing environmental exposures across GWAS, we have developed environment-adjusted meta-regression that accounts for both environmental exposures and genetic ancestry of each cohort. Building on the MR-MEGA meta-regression framework, the environment-adjusted meta-regression model, env-MR-MEGA, is constructed by adding study-level environmental covariates. This allows for detection of genetic associations by adjusting for and quantifying environmental and ancestral heterogeneity between populations.

In our extensive simulation study, we compared MR-MEGA and env-MR-MEGA in terms of the power to detect association and heterogeneity of allelic effects due to ancestry and/or environment. In our application to twelve sex-stratified cohorts of ~19,000 individuals from Africa (West, East and South Africa), env-MR-MEGA identified seven variants for LDL (Low-density lipoprotein) cholesterol that have heterogeneity beyond that explained by ancestral effects.

## Results

Accounting for different environmental exposures between cohorts, we developed an environment-adjusted meta-regression model of GWAS in which the MR-MEGA meta-regression framework[30] is adapted by adding study-level environmental covariates (Methods). This adjustment accounts for not only the heterogeneity in allelic effects that are correlated with genetic ancestry but also the heterogeneity in varied environmental exposures between GWAS.

We have designed env-MR-MEGA with two aims: (i) adjusting for environmental exposures and ancestral effects when testing if a variant is associated with a trait across GWAS; (ii) assessing whether there are environmental and/or ancestral effects that impact the association and are source(s) of allelic heterogeneity. When there is allelic heterogeneity due to environment, this suggests presence of gene-environment interactions, though this interaction is not directly assessed by env-MR-MEGA since we do not estimate/test the interaction effect. Our modelling framework allows us to assess the impact of environmental exposures on allelic heterogeneity using only study-level summaries of environmental variables, without the need for individual-level data. Based on the

environment-adjusted meta-regression model, env-MR-MEGA, we test for genetic associations and the heterogeneity in allelic effects due to genetic ancestry and environmental effects or individually from each of genetic ancestry and environment.

We compare performance of MR-MEGA and env-MR-MEGA in a series of simulations, varying environmental and ancestral effects. To simulate genotype data, we obtained allele frequencies from eight African populations: Durban Diabetes Study (DDS) and Durban Case Control Study (DCC)[35], which are two Zulu cohorts, the Uganda Genome Resource (UGD)[36] and Esan (ESN), Gambian (GWD), Luhya (LWK), Mende (MSL), and Yoruba (YRI) from Phase 3 of 1000 Genomes[37]. Based on the variants with minor allele frequencies (MAF) > 5% in all eight populations, we derived two axes of genetic variation, which suggest that the populations separate into four clusters: (i) ESN and YRI, (ii) GWD and MSL, (iii) UGD and LWK, (iv) DCC and DDS (Supplementary Fig. 1). These four clusters are consistent with geographical regions: (i) West-central Africa; (ii) West Africa; (iii) East Africa; (iv) South Africa. Then, based on these four clusters, we defined six heterogeneity scenarios for the presence of a genetic association: ancestrally homogeneous, East Africa, West Africa, West-central Africa, South Africa and non-ancestral Africa (Supplementary Table 1). For example, under the East Africa setting, populations from East Africa (UGD and LWK) are associated with a particular genetic variant, and all other populations are null at that variant.

In simulations of 16 sex-stratified GWAS from the eight African populations, we consider a range of sex-stratified models of association across our six heterogeneity scenarios, with varied sample sizes (equal medium-sized vs. unequal large-sized) and varied environmental exposure proportions across cohorts. Tag SNPs, being in LD with the causal variant, are likely to also be detected as associated with the trait by env-MR-MEGA. In the ancestrally homogeneous setting, where the causal variant has the same effect size in all populations, it is plausible that a tag SNP will have different levels of correlation with the causal variant due to different population LD patterns. In that case, env-MR-MEGA is expected to detect allelic heterogeneity due to ancestry at the tag SNP, but not the causal variant—as the causal variant is unknown in practice, we would consider the tag SNP to have allelic heterogeneity. We explore this in a simulation scenario where there is ancestral homogeneity and varying environmental exposure proportions. Finally, we apply env-MR-MEGA to LDL-cholesterol in 12 sex-stratified cohorts from Africa, with environmental covariates sex, BMI (Body Mass Index), and urban status. We evaluate and assess evidence of heterogeneity due to these environmental sources and from ancestry.

### Adjusting for a stratified environment improves power in detecting genetic associations and allelic heterogeneity

Treating sex as an environmental covariate, based on 16 sex-stratified cohorts, we examined a model of female-specific association with the trait for the six heterogeneity scenarios (Supplementary Table 1). As a baseline comparison, we consider equal-sized samples of size 1000, as well as varying sample sizes of 3000–4000 in each cohort.

Across the six heterogeneity scenarios, type 1 errors for detecting associations are well-calibrated at 0.05 (Supplementary Table 2). Amongst the six heterogeneity scenarios, under a model of female-specific effects, env-MR-MEGA gained the greatest power to detect association over MR-MEGA regardless of sample size (Fig. 1). As expected, comparing results from different sample size settings in each cohort, smaller sample sizes in female/male cohorts (1000) have substantially lower power than the larger mixed sample sizes (≥3000 in each cohort). For scenarios in which heterogeneity in female allelic effects was correlated with ancestry (East Africa, West-central Africa, West Africa, and South Africa), regardless of any sample size differences amongst cohorts, env-MR-MEGA provided the greatest power to detect association. When heterogeneity in female allelic effects across populations was homogeneous and random (non-ancestral Africa), env-MR-MEGA still attained the greatest power and had higher gains over MR-MEGA than in the four regional settings.

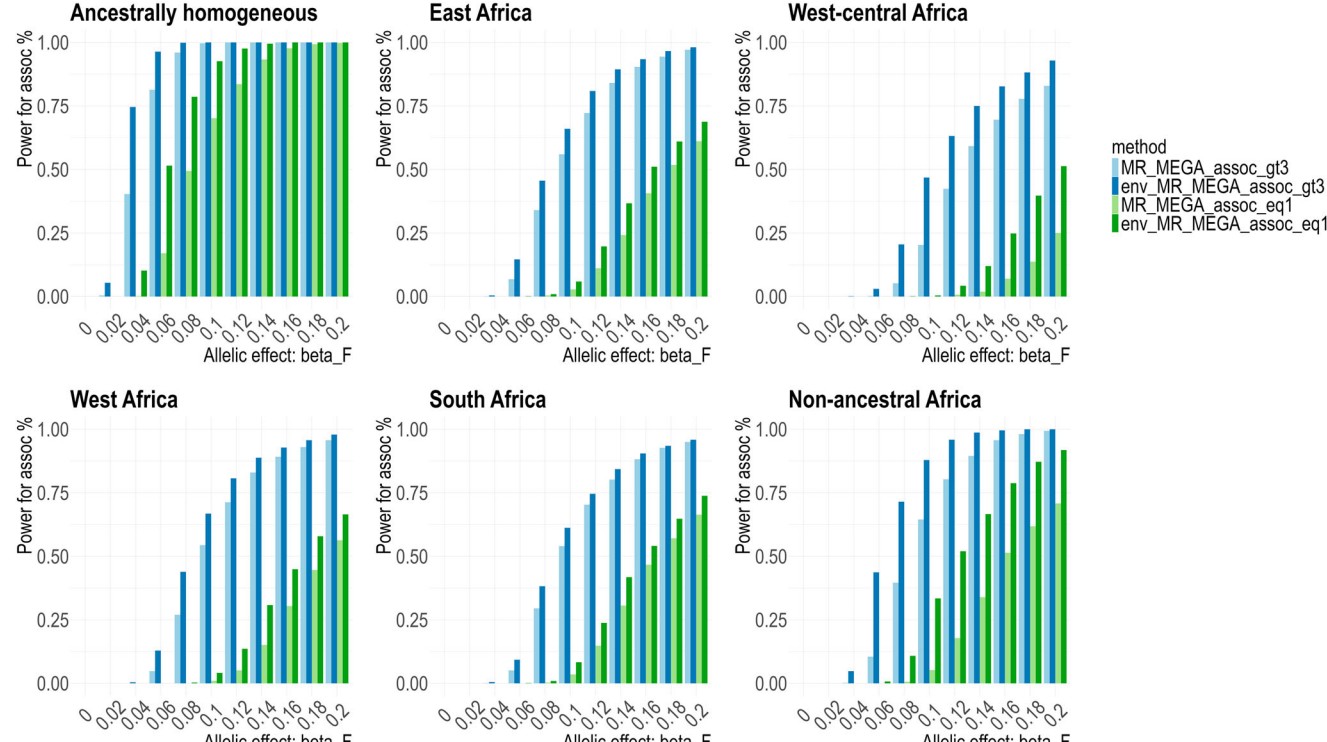

**Fig. 1 | Power to detect association by env-MR-MEGA and MR-MEGA under different ancestral heterogeneity settings.** Power to detect association by env-MR-MEGA is higher than MR-MEGA in various heterogeneity scenarios involving 16 sex-stratified cohorts from East, West, West-central and South African regions.

"MR_MEGA_assoc_gt3" and "env_MR_MEGA_assoc_gt3" correspond to the setting of unequal sample sizes (≥3000 in each cohort); "MR_MEGA_assoc_eq1" and "env_MR_MEGA_assoc_eq1" correspond to the setting of equal sample sizes (1000 in each cohort). Power was assessed at $P < 5 \times 10^{-8}$ and based on 1000 replications.

To further explore the ancestrally homogeneous scenario, we extended the homogeneity in ancestry setting to homogeneity in ancestry and sex, where both female and male cohorts in each African population share the same allelic effects. Tests for heterogeneity due to ancestry and environment by env-MR-MEGA are well-calibrated at the nominal significance level, 0.05 (Supplementary Table 3). In the model of homogeneity of allelic effects in both ancestry and sex, the type I errors for detecting (i) heterogeneity due to ancestry and environment was 0.048 (0.049 under the equal-sized setting); (ii) heterogeneity due to ancestry was 0.054 (0.040 under the equal-sized setting); (iii) heterogeneity due to environment was 0.045 (0.053 under the equal-sized setting).

Amongst the heterogeneity scenarios of East Africa, West-central Africa, West Africa, and South Africa, the power to detect heterogeneity in allelic effects due to ancestry and environment was greatest from env-MR-MEGA (Supplementary Fig. 2). As expected, the power to detect allelic heterogeneity due to ancestry was the same for env-MR-MEGA and MR-MEGA. The two settings of unequal sample size and equal sample size gave similar results for the heterogeneity tests, with the expected lower power for tests involving smaller samples (Supplementary Fig. 3). Furthermore, the power for heterogeneity due to environment was greater than that due to ancestry in the West-central Africa scenario, while the power for heterogeneity due to environment was lower than that due to ancestry in the other ancestry-specific heterogeneity scenarios (East Africa, West Africa, and South Africa). This suggests that the allelic heterogeneity in the West-central Africa scenario shows lower correlations with ancestry compared to sex effect, unlike the other ancestry-specific scenarios, East Africa, West Africa, and South Africa, where allelic heterogeneity shows a stronger correlation with ancestry. Possibly this is because the axes of genetic variation for West-central Africa are between those for West Africa and East Africa (Supplementary Fig. 1).

For the ancestrally homogeneous scenario, the power to detect allelic heterogeneity due to ancestry from both env-MR-MEGA and MR-MEGA attained the nominal significance threshold, $\alpha = 0.05$ (Supplementary Figs. 2 and 3). In addition, the power to detect allelic heterogeneity due to environmental factors was slightly higher than the power for allelic heterogeneity due to ancestry and environmental effects and it was notably greater than the power for heterogeneity due to ancestry alone. This is because the heterogeneity in allelic effects across sex-stratified cohorts was correlated with sex alone. For the non-ancestral Africa scenario, the power to detect heterogeneity was noticeably lower for ancestral effects only compared to environmental effects because of the randomness in assigned effects among population groups.

## Adjusting for environment proportions improves power in detecting genetic associations and allelic heterogeneity

To investigate the impact of an environmental effect, we considered a more complicated setting in which the heterogeneity in allelic effects is correlated with the environmental factor instead of sex alone. Here, smoking status was treated as the environmental covariate, and the proportion of smokers varied across the eight populations, including the sex-stratified 16 male/female cohorts. The smoking proportions vary across male and female cohorts and across different ancestries. To explore a broader range of scenarios, we considered two smoking patterns: (i) small differences in smoking proportions between males and females but large differences in smoking proportions between populations in the same African region (Supplementary Table 4); (ii) large differences in smoking proportions between male and females but minor differences in smoking proportions between populations in the same African region (Supplementary Table 5). Additionally, for each smoker pattern, we considered three settings: (a) no difference: female and male cohorts in the same population shared the same smoking proportions; (b) same direction: smoking proportions in female cohorts were always lower than that in male cohorts; (c) mixed direction: among the eight populations, not all female smoking proportions were lower than that in male cohorts

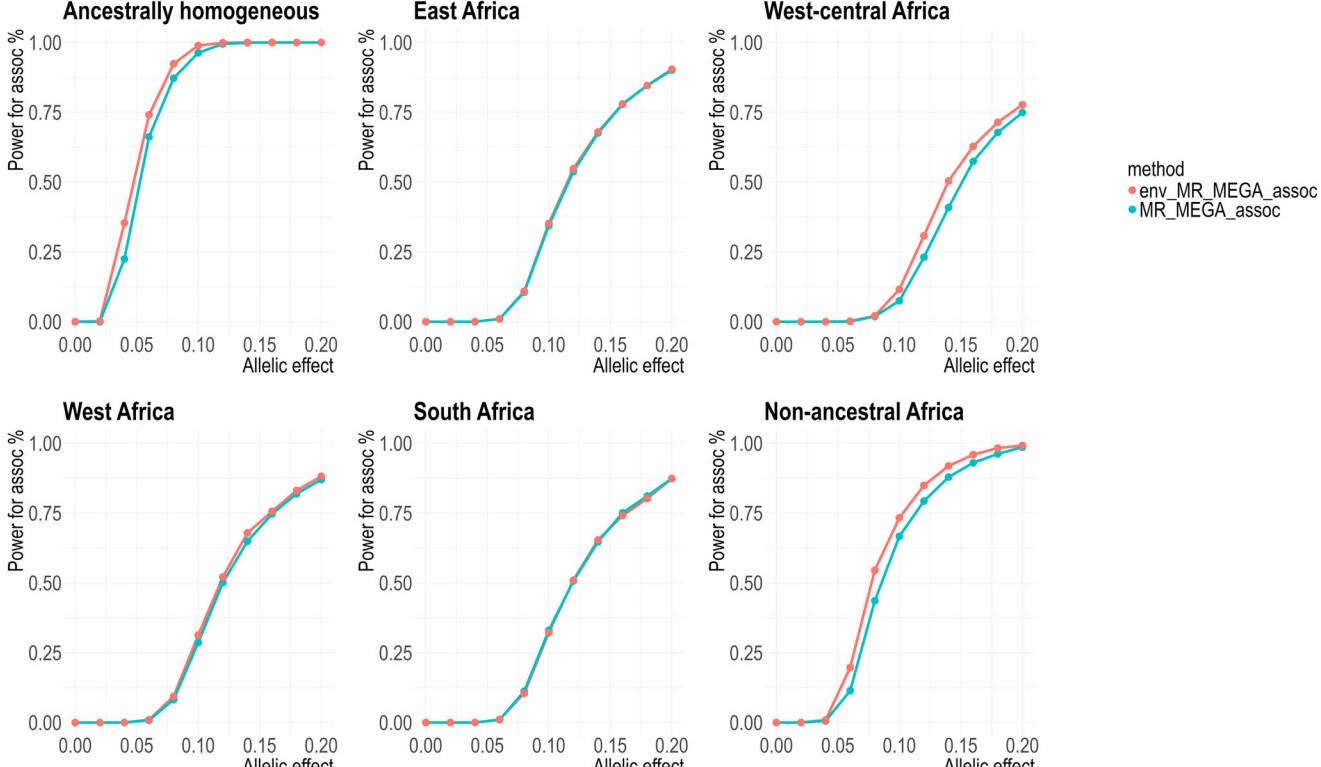

**Fig. 2 | Power to detect association by env-MR-MEGA and MR-MEGA under different ancestral heterogeneity settings with similar smoking proportions between male and female cohorts.** Across six heterogeneity scenarios involving 16 sex-stratified cohorts, where there are minor random reductions or increases in smoking proportions between male and female cohorts (mixed direction), env-MR-MEGA exhibits higher power in detecting association compared to MR-MEGA, especially in the ancestrally homogeneous and non-ancestral Africa scenarios. "env_MR_MEGA_assoc" and "MR_MEGA_assoc" refer to the power to detect association obtained from env-MR-MEGA and MR-MEGA. Power was assessed at $P < 5 \times 10^{-8}$ and based on 1000 replications with unequal sample sizes ($\geq$3000 in each cohort).

(randomly selected populations where female smoker proportion is higher than that for males). Consequently, through a range of simulations under these settings, it is demonstrated that env-MR-MEGA had the highest gains in power over MR-MEGA when there was higher variability in smoking proportions between the cohorts.

In the first smoking pattern, where male and female cohorts from the same population share identical smoking proportions or varied smoking proportions with small differences (Supplementary Table 4), env-MR-MEGA gained the greatest power to detect association over MR-MEGA, particularly in the ancestrally homogeneous, West-central Africa and non-ancestral Africa scenarios (Fig. 2, Supplementary Figs. 4 and 5). For the East Africa and West Africa scenarios, env-MR-MEGA gained slightly higher power than MR-MEGA, whereas, in the South Africa scenario, the power to detect association was nearly equivalent for both env-MR-MEGA and MR-MEGA. It can be explained that when heterogeneity in allelic effects is more strongly correlated with ancestry than with smoking exposures, especially in the South Africa scenario where the two Zulu cohorts (DCC and DDS) are highly similar to each other (Supplementary Fig. 1), the power for association is more significantly influenced by ancestry compared to smoking exposures. Consequently, env-MR-MEGA and MR-MEGA have similar powers for association in the South Africa scenario. Additionally, under this smoking pattern, there was no notable difference in power of association in each heterogeneity scenario between the three settings for the smoking proportion differences between males and females, which include no difference, mixed direction and same direction (Supplementary Figs. 4 and 5).

As the disparity in smoking proportions between males and females increased and the variability in smoking proportions between African regions diminished (Supplementary Table 5), compared to the first smoking proportion pattern (Fig. 2, Supplementary Figs. 4 and 5), the difference in power to detect association between env-MR-MEGA and MR-MEGA in the

same direction setting became more apparent (Fig. 3). Compared with the same direction setting, where all female cohorts have lower smoking proportions, the mixed direction setting includes South Africa and West-central Africa scenarios where one female cohort has a higher smoking proportion. Therefore, in the South Africa and West-central Africa scenarios, the power for association attained from env-MR-MEGA in the same direction setting was notably higher than that in the mixed direction setting. However, the power for association from MR-MEGA varied less between same direction and mixed direction settings, compared to the env-MR-MEGA power differences between these two settings.

In the first pattern, where the smoking proportions varied slightly between male and female cohorts (Supplementary Table 4), the power to detect heterogeneity due to ancestry and/or environment does not vary dramatically across the three settings for smoking proportions: no difference, same direction and mixed direction (Supplementary Figs. 6, 7, and 8). For the ancestrally homogeneous, West-central Africa and non-ancestral Africa scenarios, where heterogeneity in allelic effect has lower correlation with ancestry compared to smoking exposures, the power for allelic heterogeneity due to ancestry and environment from env-MR-MEGA was notably greater than that due to ancestry from MR-MEGA. For the East Africa and West Africa scenarios, where heterogeneity in allelic effects is moderately correlated with ancestry and smoking exposures, there was not much difference between the power for heterogeneity due to ancestry and the power for heterogeneity due to ancestry and environment. For the South Africa scenario, where the heterogeneity in allelic effect is more strongly correlated with South Africa ancestry compared to smoking exposures, the power for allelic heterogeneity due to ancestry attained from both env-MR-MEGA and MR-MEGA is moderately greater than that due to ancestry and environment. For the no difference setting and mixed direction setting in the South Africa scenario, where the differences in smoking proportions

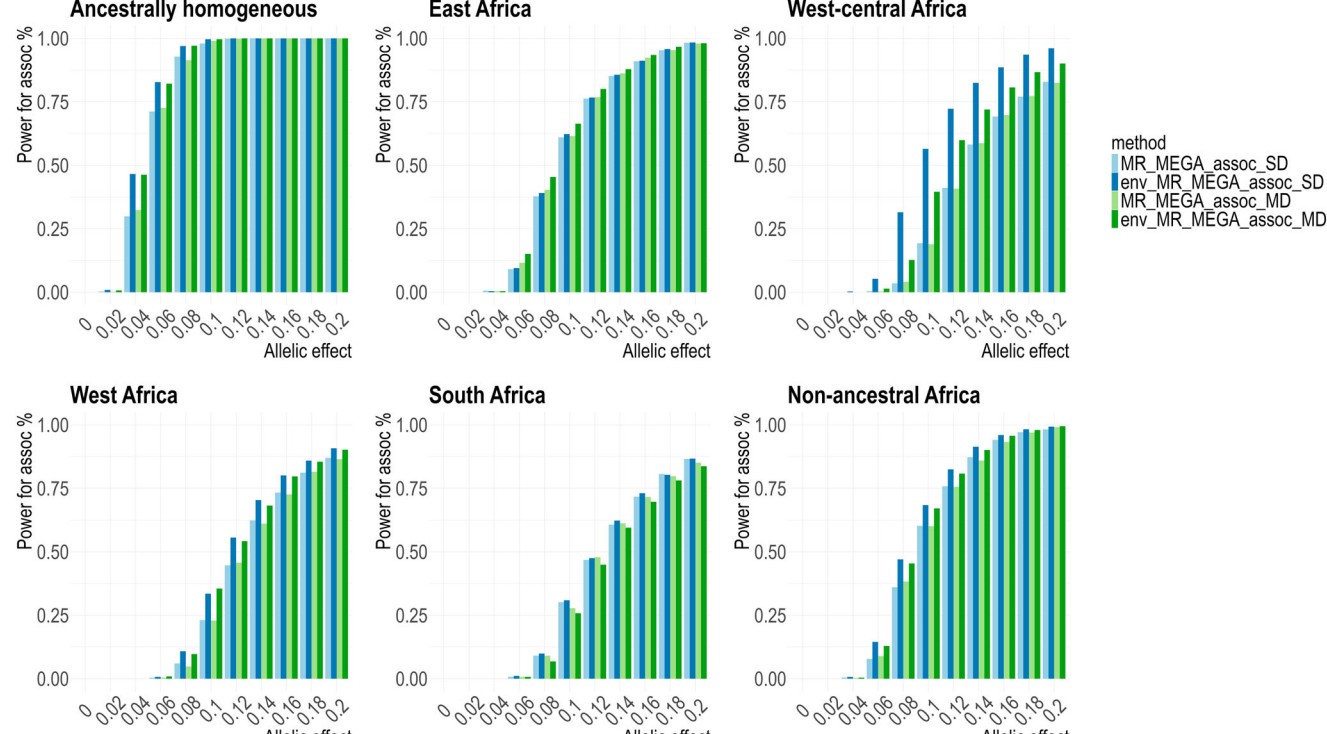

**Fig. 3 | Power for association tests by env-MR-MEGA and MR-MEGA when female and male cohorts have different smoking patterns, under different ancestral heterogeneity settings.** When smoking proportions clearly differed between male and female cohorts, env-MR-MEGA gained greater power to detect association compared to MR-MEGA. In the mixed direction pattern, where not all male cohorts share high smoking proportions, both West-central Africa and South Africa scenarios contain one male cohort with a lower smoking proportion. Within West-central Africa and South Africa scenarios, comparing the same direction and the mixed direction patterns, the gains in power to detect association by env-MR-MEGA in the same direction pattern exceeded those in the mixed direction pattern. "MR_MEGA_assoc_SD" and "env_MR_MEGA_assoc_SD" correspond to the same direction pattern, while "MR_MEGA_assoc_MD" and "env_MR_MEGA_assoc_MD" correspond to the mixed direction pattern. Power was assessed at $P < 5 \times 10^{-8}$ and based on 1000 replications with unequal sample sizes (≥3000 in each cohort).

between male and females are not consistent with sex effects, the power for allelic heterogeneity due to ancestry is noticeably greater than that due to ancestry and environment when the allelic effect of the causal variant varied between 0.04 and 0.06. For the same direction setting, where female cohorts have lower smoking proportions, detection of allelic heterogeneity due to ancestry and environment had clear gains in power over ancestry alone (MR-MEGA and env-MR-MEGA) when there is a moderate allelic effect, around 0.06 (Supplementary Fig. 9).

For the scenarios where heterogeneity in allelic effect is correlated with ancestry and smoking exposures (East Africa, West Africa, West-central Africa and South Africa), the power for allelic heterogeneity due to environment from env-MR-MEGA was the lowest amongst all heterogeneity tests in the first smoking pattern (Supplementary Table 4, Supplementary Figs. 6, 7, and 8). As the difference in smoking proportions between female and male cohorts increased and variation in smoking proportions between ancestries diminished (Supplementary Table 5), the power for heterogeneity due to the environment increased significantly, particularly for the West-central Africa scenario (Supplementary Figs. 10 and 11).

In the mixed direction setting, where one of the two populations had a higher smoking proportion for females than males in the South Africa and West-central Africa regions (Supplementary Table 5), there is a loss in power to detect environmental heterogeneity over the setting where both populations in a region have the same smoker pattern between sexes (Supplementary Figs. 10 and 11). Also, we note that in the mixed-direction setting, we randomly selected populations to have a higher proportion of female smokers than male. This resulted in another version of the "same" setting for the East Africa region, where both populations have a higher proportion of female smokers. Consequently, the East Africa region has a higher power to detect environmental heterogeneity in the mixed

simulations than in the same setting simulations, where female smoker proportions were lower in both populations. Finally, in the mixed direction setting, the West Africa female smoker proportions remain identical to the same direction setting. However, there are slight differences between the power estimates between these two settings for West Africa. Although the data generated under the same and mixed settings use the same smoking proportions for West Africa and there is only a smoking effect for West Africa populations, the observed difference arises from our application of env-MR-MEGA. In applying env-MR-MEGA, we use the smoking proportions from all populations under the same or mixed setting, for which there are several slight differences due to setting the females to have a higher smoker proportion in some populations. As a result, the power to detect environmental heterogeneity is similar in the same and mixed settings. In contrast, the power to detect ancestral heterogeneity is slightly lower in the same direction setting compared to the mixed setting. This is explained by the smaller chi-squared statistic that results from similar female smoking proportions in the same direction setting, compared to the larger differences in the mixed setting.

Additionally, we investigated the performance of env-MR-MEGA at a tag SNP (single nucleotide polymorphism) of a causal variant, where the LD between the causal variant and tag SNP is similar in populations from the same region and varies between regions (Methods). In the ancestrally homogeneous scenario, we replicated the smoking proportions across 16 sex-stratified cohorts in the second smoking pattern (b) in the same direction setting (Supplementary Table 5). Sequentially, we generated 1000 replicates of genotype data for a causal variant and its correlated genetic variant. We then applied env-MR-MEGA to each of the 1000 causal variants and 1000 tag SNPs to estimate the power for association and heterogeneity due to ancestry and/or environment at causal variants and at tag SNPs.

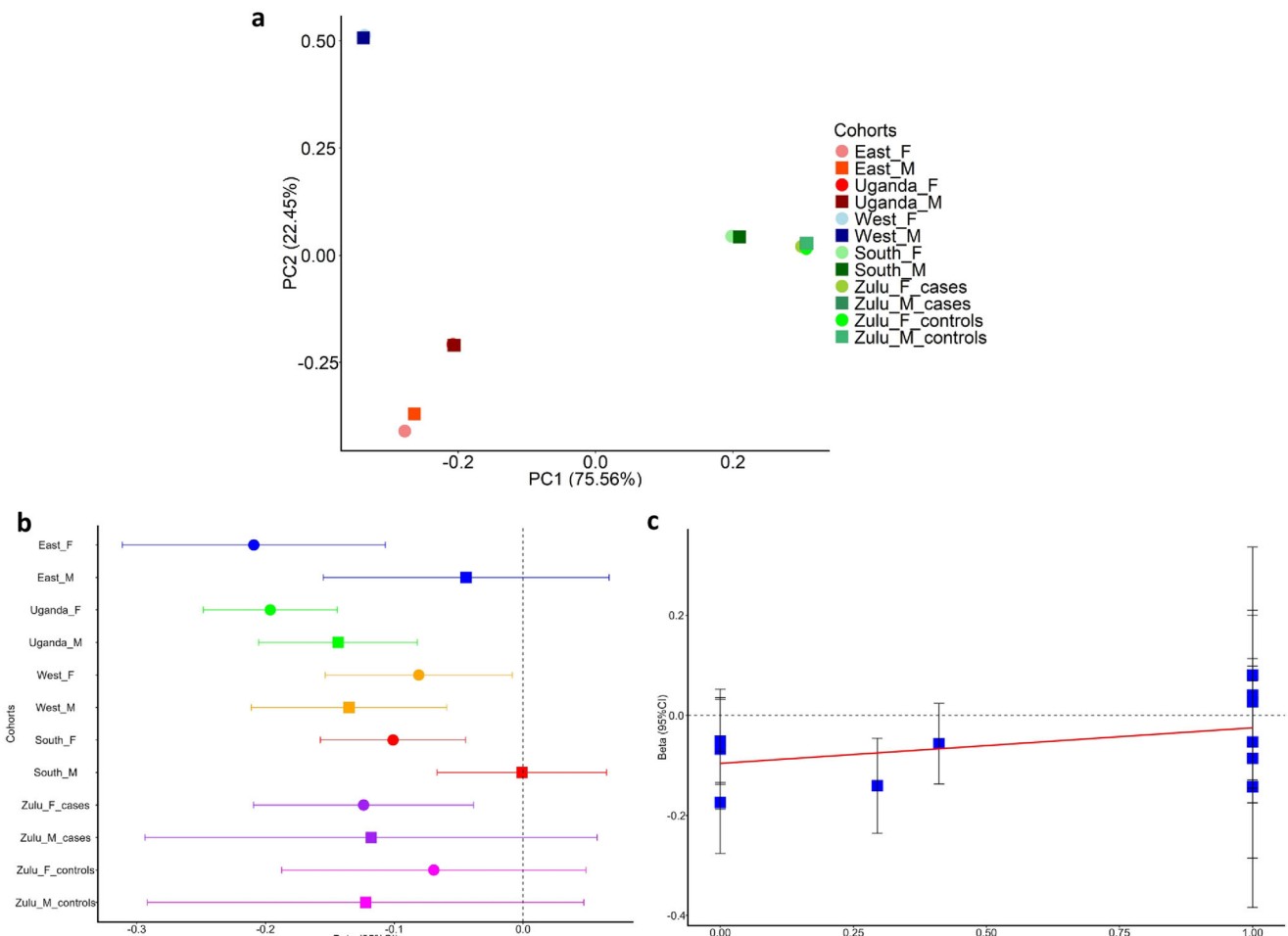

**Fig. 4 | Axes of genetic variation and allelic heterogeneity due to environmental factors across twelve sex-stratified African cohorts. a** Axes of genetic variation: axes of genetic variation showing separation into three distinct clusters representing cohorts from West Africa (West_M & West_F), East Africa (East_M & East_F and Uganda_M & Uganda_F), and South Africa (South_M & South_F, controls for Zulu_M & Zulu_F and cases for Zulu_M and Zulu_F). **b** heterogeneity due to sex with the female cohorts (for Zulu_cases, South, Uganda and East) exhibiting greater effect sizes for rs12740374 than males. **c** heterogeneity due to urban status with effect size for rs3735018 increasing towards urban residents' cohorts. In (**b**) and (**c**) 95% confidence intervals are provided as error bars for each effect estimate. Source data for b and c are available in Supplementary Data 1.3.

It is demonstrated that, when the allelic effect varied between 0.02 and 0.1, the power for association at causal variants is notably greater than that from genetic variants that are in LD with causal variants (different LD amongst populations) (Supplementary Fig. 12). Likewise, at low-moderate effect sizes of 0.02 to 0.1, power to detect allelic heterogeneity due to ancestry and environment or due to only environment, is notably higher at the causal variants compared to the tag SNPs. As expected, the power for heterogeneity due to ancestry attained the nominal significance level for the causal variants. However, at the tag SNPs, the power for ancestral heterogeneity increased with the size of the allelic effect. This is due to the lower correlation with the causal variant in populations from West Africa and South Africa.

**Identification of sources of genetic heterogeneity in African cohorts**

We applied the env-MR-MEGA and MR-MEGA methods to the summary statistics of twelve sex-stratified African GWAS (representing West Africa, East Africa and South Africa) for LDL-cholesterol using sex, mean BMI, and the proportion of study participants with urban status as environmental variables (Methods, Supplementary Data 1.1). We used these variables individually and in pairs as environmental factors. We considered 45 SNPs that have previously attained genome-wide significance for LDL-cholesterol in the meta-analysis of the substantially larger African American GWAS

from the Global Lipids Genetics Consortium (GLGC)[38] and are present in our data (Supplementary Data 1.2). For these SNPs, we used env-MR-MEGA and MR-MEGA to first test for association in our African cohorts using a Bonferroni-corrected threshold for association of 0.001 (0.05/45) and a suggestive threshold of 0.005. Amongst the SNPs that env-MR-MEGA/MR-MEGA identified as associated in our data, we investigated the contribution of ancestry and environmental variables (sex, BMI, and urban environment) to heterogeneity in allelic effects using a nominal threshold of 0.05 (as detection of association has been adjusted for multiple tests).

We note that testing for heterogeneity at variants that have evidence of association does not inflate the type 1 error of these heterogeneity tests. This is because we include the ancestral and environmental covariates in our model to test for association, without any previous testing for significance of these covariates. If we had first tested the sources of heterogeneity and then only included in our model the sources that have evidence of heterogeneity, then we would encounter the problem of forking paths and consequently, inflated type 1 error. In the forking path problem, variables are first tested and then included in the model if they are found to have a significant effect, assuming that otherwise such variable(s) would not have been included in the model[39].

The axes of genetic variation show three main clusters, coinciding with East Africa (AWI-Gen East, Uganda), West Africa (AWI-Gen West), and South Africa (AWI-Gen South, Zulu) (Fig. 4a). Nine SNPs (Table 1) showed

**Table 1 | Env-MR-MEGA p-values for heterogeneity due to ancestry and environmental variables**

| SNP (CHR:POS) | Environment-adjusted MR-MEGA | | | | | | | | | | | |
| | Sex | | | BMI | | | Sex + BMI | | | Urban status | | |
| | Ancestral + environmental | Environmental | Ancestral | Ancestral + environmental | Environmental | Ancestral | Ancestral + environmental | Environmental | Ancestral | Ancestral + environmental | Environmental | Ancestral |
|---|---|---|---|---|---|---|---|---|---|---|---|---|
| rs28362286 (1:55529215) | 0.833 | 0.465 | 0.800 | 0.930 | 0.731 | 0.807 | 0.907 | 0.709 | 0.995 | 0.199 | 0.038** | 0.531 |
| rs12740374 (1:109817590) | 0.001** | 0.026** | 0.003** | 0.002** | 0.055 | 0.001** | 0.003** | 0.069 | 0.015** | 0.011** | 0.840 | 0.007** |
| rs1712248 (2:21385778) | 0.010** | 0.053 | 0.033** | 0.031** | 0.259 | 0.341 | 0.023** | 0.153 | 0.166 | 0.047** | 0.572 | 0.022** |
| rs11884143 (2:24712930) | 0.003* | 0.521 | 0.001* | 0.003* | 0.474 | 0.001* | 0.003* | 0.259 | 0.001* | <0.001* | 0.027* | 0.005* |
| rs4245791 (2:44074431) | 0.076 | 0.839 | 0.032* | 0.018* | 0.072 | 0.268 | 0.010* | 0.038* | 0.197 | 0.022* | 0.096 | 0.054 |
| rs3735018 (7:137562744) | 0.795 | 0.587 | 0.685 | 0.468 | 0.178 | 0.395 | 0.610 | 0.374 | 0.370 | 0.158 | 0.035** | 0.180 |
| rs3808607 (8:59412924) | 0.015* | 0.565 | 0.007* | 0.006* | 0.129 | 0.006* | 0.013* | 0.282 | 0.005* | 0.001* | 0.019* | 0.043* |
| rs138294113 (19:11191729) | 0.070 | 0.270 | 0.044** | 0.113 | 0.724 | 0.181 | 0.121 | 0.484 | 0.596 | 0.109 | 0.644 | 0.068 |
| rs7412 (19:45412079) | 0.017** | 0.133 | 0.017** | 0.029** | 0.308 | 0.013** | 0.037** | 0.323 | 0.050 | 0.004** | 0.021** | 0.003* |

The env-MR-MEGA heterogeneity $p$-values are provided at variants with evidence of ancestral and/or environmental heterogeneity in 12 sex-stratified African cohorts. At GLGC variants that meet the Bonferroni-adjusted threshold (0.001) of genetic association in our data, we highlight heterogeneity $p$-values < 0.05 by presenting them with two asterisks. At GLGC variants that meet the suggestive threshold (0.005) of genetic association, we highlight heterogeneity $p$-values < 0.05 by presenting them with one asterisk. Ancestral and/or environmental (sex, BMI (body mass index), sex + BMI, urban status) heterogeneity $p$-values were generated using the env-MR-MEGA methods.
SNP single nucleotide polymorphism, *CHR:POS* chromosome and position of each SNP; Ancestral: heterogeneity due to ancestry only; Environmental: heterogeneity due to environment only; Ancestral+environmental: heterogeneity due to ancestry and environment.

allelic heterogeneity (*p*-value < 0.05) due to ancestry, environment only (sex, BMI, and urban status) and a combination of both ancestry and environmental variables.

Ancestral heterogeneity is detected by both MR-MEGA and env-MR-MEGA at five variants (rs12740374, rs1712248, rs11884143, rs4245791, rs7412), and env-MR-MEGA detects ancestral heterogeneity at an additional two variants (rs138294113, rs3808607) (Supplementary Data 1.2). In addition, env-MR-MEGA shows evidence of heterogeneity due to ancestry and environment at six of these variants; at rs138294113 env-MR-MEGA only detects ancestral heterogeneity. Our env-MR-MEGA method detected allelic heterogeneity involving environmental effects in seven SNPs: one SNP driven by sex (rs12740374), one SNP jointly driven by combined sex and BMI (rs4245791), and five SNPs driven by urban status (rs28362286, rs11884143, rs3735018, rs3808607, and rs7412). rs28362286 and rs7412 were genome-wide significant SNPs for LDL-cholesterol in our twelve African cohorts and overlap with the GLGC meta-analysis results[38] (Table 1; Supplementary Data 1.2). Female cohorts from East Africa are most strongly associated with decreased LDL-cholesterol (Fig. 4b, Supplementary Data 1.2), while those residing in urban areas are most strongly associated with increased LDL-cholesterol (Fig. 4c, Supplementary Data 1.3).

## Discussion

We have developed an environment-adjusted meta-regression of GWAS, env-MR-MEGA, utilising GWAS summary statistics, along with environmental exposures and ancestry. The model is built upon the framework of MR-MEGA[30], treating the allelic effects as a function of environmental exposures in addition to genetic ancestry. Through a series of simulations, we demonstrate that env-MR-MEGA outperforms MR-MEGA in terms of power to detect association when there is indeed an effect of the environmental exposure on heterogeneity, as well as the power to detect heterogeneity that is due to various sources, i.e. genetic ancestry and/or environmental factors.

The framework of env-MR-MEGA not only accounts for ancestry obtained by deriving axes of genetic variation via multi-dimensional scaling but also study-level environmental impacts obtained by taking the mean or proportion of the individual-level environmental data within each cohort. Our simulation studies and African data application focused on cohorts from different African regions, which include East Africa, West Africa, and South Africa. In our analyses, we have focused on the two axes of genetic variation and illustrated that the two axes of genetic variation are sufficient to distinguish these cohorts within the African continent (Supplementary Fig. 1, Fig. 4a). The environmental exposures we considered are the sex-stratification binary variable, the mean BMI within each cohort, and the study-level urban status proportions. By incorporating genetic variation axes and study-level environmental exposures as covariates in the meta-regression model, env-MR-MEGA can detect heterogeneity in allelic effects attributed to ancestry and environmental factors, respectively.

When the sex-stratification variable is the only environmental exposure and allelic effects are correlated with sex and ancestry, our simulations show that, across all scenarios, env-MR-MEGA gains the greatest power to detect association over MR-MEGA. Additionally, as the correlation between allelic heterogeneity and ancestry decreases, such as in the ancestrally homogeneous scenario and the non-ancestral scenario, the test for allelic heterogeneity due to environmental effects gains more power than that due to ancestry (Supplementary Fig. 2). Applying env-MR-MEGA to association studies of LDL-cholesterol highlights notable evidence of heterogeneity in sex, which was observed at rs12740374 (Supplementary Data 1.2).

Our simulation results demonstrate that when the heterogeneity in allelic effects has a lower correlation with ancestry compared to the environmental exposures, such as ancestrally homogeneous, west-central Africa and non-ancestral Africa scenarios, env-MR-MEGA gains noticeably higher power to detect association over MR-MEGA (Fig. 2, Supplementary Figs. 4 and 5). This improvement in power is attributed to the weak correlation between allelic heterogeneity and ancestry. Specifically, as the correlation between allelic heterogeneity and ancestry weakens, the power

for heterogeneity due to ancestry and environment from env-MR-MEGA became notably greater than that due to ancestry alone from MR-MEGA.

In our application to GWAS meta-analysis of LDL-cholesterol across African GWAS, BMI and urban status are treated as environmental exposures in addition to sex-stratification. Our assessment of heterogeneity of genome-wide significant SNPs in the GLGC study revealed seven SNPs with evidence of environmental heterogeneity and two SNPs with heterogeneity due to ancestry only. Complex traits are influenced by a combination of genetic and environmental factors; thus, genome-wide interaction studies (GWIS) have been modelled to understand how genetic variants modify the effect of environmental exposures on a disease or trait[40,41]. Our method, env-MR-MEGA, differs from GWIS, as it specifically targets environmental heterogeneity to reduce confounding and identify genetic association signals. In a recent investigation into gene-environment interactions involving LDL-cholesterol, independent loci near *CELSR2*: rs12740374, *APOB*: rs1712248, *APOE*: rs7412 and *LDLR/SMARCA4*: rs138294113 which we identified in our application, have been shown to interact with two environmental factors (cigarette smoking and alcohol consumption) and contributed to heterogeneity in allelic effects across different populations[41]. Thus, underscoring the consistency and environmental influences on these genetic association signals across different methodological approaches.

Compared to MR-MEGA, the key advantage of env-MR-MEGA is that the environment-adjusted meta-regression model accounts for heterogeneity in allelic effects that can be attributed to ancestry and environmental factors instead of ancestry alone. When there is heterogeneity in allelic effects that is not correlated with ancestry, we can quantify the contribution of environmental effects to the heterogeneity of allelic effects between GWAS. Consequently, env-MR-MEGA offers a more comprehensive approach to the discovery across GWAS from diverse populations. We expect that env-MR-MEGA will improve PRS prediction in African populations, which we plan to implement in future work. With the increasing availability of GWAS from more diverse populations, an efficient statistical methodology that will account for heterogeneity in meta-analysis of GWAS, such as env-MR-MEGA, shows great promise for future improvements in our understanding of the genetic basis of complex human traits.

## Methods

Following MR-MEGA[30], consider *K* sets of GWAS summary statistics of a complex trait from *K* cohorts and assume that allelic effects for all cohorts are aligned to the same reference allele at each variant. For the *k*th study, we denote by $p_{kj}$ the reference allele frequency of the *j*th SNP in the *k*th cohort. We derive a matrix of pairwise Euclidean distances between cohorts from a subset of autosomal variants, denoted $D = [d_{kk'}]$. More specifically, each component $d_{kk'}$ of the matrix $D$ is given by $d_{kk'} = \frac{\sum_j I_j (p_{kj} - p_{k'j})^2}{\sum_j I_j}$, where $I_j$ is a binary variable indicating whether the *j*th variant is in the subset of SNPs used for calculating the distance matrix.

In the procedure of generating the subset to derive the distance matrix, we follow the strategy of MR-MEGA[30]. First, the autosomal variants from the reference panel with minor allele frequency (MAF) > 5% in all cohorts are retained and are divided into 1 Mb bins. Then we randomly select one variant from each bin and aggregate them to calculate the pairwise Euclidean distances.

We denote *T* axes of genetic variation across *K* cohorts by $x_t = (x_{1t}, x_{2t}, \ldots, x_{Kt})$, $t = 1, \ldots, T$, which are derived from multidimensional scaling of the distance matrix. Considering that the environmental exposures could be specific to each cohort, we construct the cohort-level environmental covariates by taking the mean or proportion of the individual-level environmental data within each cohort. Here we denote *S* cohort-level environmental covariates across K GWAS by $z_s = (z_{1s}, z_{2s}, \ldots, z_{Ks})$, $s = 1, \ldots, S$. It is noted that the number of axes of genetic variation and environmental covariates permitted is limited by the number of cohorts, *K*. In particular, the number of axes of genetic

variation, $T$, and the number of environmental covariates, $S$, should satisfy $T + S < K - 2$.

Following the notation for MR-MEGA, we denote the estimated allelic effect and the corresponding variance of the $k$th cohort at the $j$th variant by $b_{kj}$ and $v_{kj}$. For the $j$th variant, our environment-adjusted meta regression model is given by

$$E\left(b_{kj}\right) = \beta_{0j} + \sum_{t=1}^{T} \beta_{tj} x_{kt} + \sum_{s=1}^{S} \alpha_{sj} z_{ks} \qquad (1)$$

where $\beta_{tj}$ represents the effect of the $t$th axis of genetic variation for the $j$th variant and $\alpha_{sj}$ represents the effect of the $s$th environmental covariate for the $j$th variant. In the model, $\beta_{0j}$ is the intercept for the $j$th variant and can be interpreted as the expected allelic effect of variant $j$ when all axes of genetic variation and environmental covariates are zero.

To test the null hypothesis of no association of the $j$th variant across all cohorts, we compare the null model with $\beta_{0j} = \beta_{1j} = \ldots = \beta_{Tj} = \alpha_{1j} = \ldots = \alpha_{Sj} = 0$ in model (1) to that without constrained parameters. The resulting test statistic for the $j$th variant has chi-square distribution with $(T + S + 1)$ degrees of freedom. When there is evidence of a genetic association, we may then test for the presence of allelic heterogeneity due to ancestral and environmental effects, as well as heterogeneity from each source alone.

To test for ancestral and environmental heterogeneity, we compare the model with $\beta_{1j} = \ldots = \beta_{Tj} = \alpha_{1j} = \ldots = \alpha_{Sj} = 0$ in model (1) to that without constrained parameters. The corresponding test statistic has an approximate chi-square distribution with $(T + S)$ degrees of freedom. In a similar manner, we also test for heterogeneity due to ancestry and environment separately. For ancestral heterogeneity in allelic effects, the test statistic is obtained by comparing the model with $\beta_{1j} = \ldots = \beta_{Tj} = 0$ to that without constrained parameters, which follows an approximate chi-square distribution with $T$ degrees of freedom. In the same manner, we test for heterogeneity in environmental effects by comparing the model with $\alpha_{1j} = \ldots = \alpha_{Sj} = 0$ to the model without constrained parameters, with the test statistic following the approximate chi-square distribution with $S$ degrees of freedom. Finally, after accounting for ancestry and environment, we can test residual heterogeneity in allelic effects by the deviance of the model (1) with unconstrained parameters. The corresponding resulting test statistic has an approximate chi-square distribution with $(K - S - T - 1)$ degrees of freedom.

### Simulation study design

We conducted extensive simulation studies to compare the performance of env-MR-MEGA and MR-MEGA in terms of statistical power to detect genetic association and heterogeneity due to environment and ancestry across genetically diverse populations from Africa. Here, assuming Hardy–Weinberg equilibrium (HWE), we simulated genotypes from the allele frequencies of eight populations from Africa: two Zulu cohorts from southern Africa, DDS and DCC[35], the Uganda Genome Resource[36] and Esan (ESN), Gambian (GWD), Luhya (LWK), Mende (MSL), and Yoruba (YRI) from Phase 3 of 1000 Genomes[37] (Supplementary Table 1).

We considered two general settings in our simulations of 16 cohorts, sex-stratified across the eight populations from Africa: (i) heterogeneity due to genetic ancestry and sex (different model for each sex); (ii) heterogeneity due to genetic ancestry and smoking status (same model for both sexes).

In our simulations where allelic effects are correlated with ancestry and sex, we set: $Y = \varepsilon$ for males and $Y = \beta \times g + \varepsilon$ for females, where $g$ denotes the genotype for one genetic variant which would be coded 0, 1 or 2 representing the number of copies of the minor allele, $\beta$ is the effect size for the genetic variant and $\varepsilon \sim N(0, sd)$, $sd = 0.45$. Here, we treat sex as an environmental covariate that acts as an indicator variable since we stratify population groups by sex. In our simulations where there is no sex effect and heterogeneity in allelic effects is correlated with ancestry and smoking status, we set: $Y = \varepsilon$ for non-smokers; $Y = \beta \times g + \varepsilon$ for smokers, where $\varepsilon \sim N(0, sd)$, $sd = 0.45$.

We considered six heterogeneity scenarios across population groups, parameterised in terms of allelic effects $\beta$ in each population group (Supplementary Table 1). These heterogeneity scenarios could be summarised into three groups: ancestrally homogeneous, ancestry-specific, and non-ancestral. For the ancestrally homogeneous setting, the allelic effect is the same across all African sub-populations (i.e. no ancestral heterogeneity). In the population-specific setting, the allelic effect is specific to sub-populations in the specified geographic region (East Africa, West-central Africa, West Africa, or South Africa). Finally, for the non-ancestral setting, the allelic effect is specific to one sub-population in each African region, i.e. heterogeneity within each African region. Details of specific simulation studies follow.

### Simulation study 1: heterogeneity in allelic effect is correlated with sex and population groups

Considering an environment-adjusted meta-analysis of eight GWAS of a quantitative trait, we initially assumed that the eight sub-populations from Africa - ESN, YRI, GWD, MSL, DCC, DDS, LWK and UGD - have unequal sample sizes of 8000, 6500, 7000, 8000, 6000, 6000, 6500 and 8000, respectively. We also stratified each sub-population by sex. For simplicity, we assumed equal sample sizes for male and female cohorts from the same sub-population. We considered a model of female-specific effect ($\beta = 0$ in male cohorts) for the above-mentioned six heterogeneity scenarios (Supplementary Table 1). Additionally, we considered the homogeneity in sex and ancestry scenario, where male and female cohorts share the same model $Y = \beta \times g + \varepsilon$ across all cohorts, to assess the type 1 error of env-MR-MEGA. In env-MR-MEGA, the environmental covariate for sex is a binary vector, each component consisting of 1 for female and 0 for male.

To evaluate the power and type 1 errors for each heterogeneity scenario, in a series of simulations, we first selected 1000 causal variants with MAF > 1%. The genotype data of these causal variants were simulated under the HWE assumption in all populations, using their population-specific allele frequencies. Under each allelic effect $\beta$, the quantitative traits were simulated from each of the 1000 causal variants' genotypes. In addition, within each setting of the effect size $\beta$, we set a sequence of seeds for generation of the error term in our simulation of Y. This same sequence of seeds was used for each setting of $\beta$. As a baseline power comparison between the two methods, we considered equal-sized cohorts (1000 in each female/male cohort) and then unequal sample sizes.

### Simulation study 2: the heterogeneity in allelic effect is correlated with the environmental covariate and population groups

To investigate the impact of the environmental effect, we considered the setting in which the heterogeneity in allelic effect is correlated with the environmental factor. Here, smoking status was treated as the environmental covariate, and the proportion of smokers varied across the 8 sub-populations, including the sex-stratified 16 male/female cohorts. We replicated the procedure outlined in the sex-stratified simulations to simulate genotype data; sample sizes and genotype data of 1000 causal variants were consistent with those in the sex-stratified simulations. We considered a model of smoker-specific association with the trait, where individuals who are smokers within each cohort have a genetic effect, while non-smokers have no genetic effect. The smoker-specific allelic effect patterns follow the six heterogeneity scenarios previously described for sex-specific allelic effect simulations (Supplementary Table 1); the same pattern is used for males and females and the effect size varies from 0 to 0.2 with increments of 0.02. The environmental covariate is a vector of smoking proportions for the corresponding female and male cohorts in the associated environment-adjusted meta-regression model.

We considered three different settings for the difference in smoking proportions between female and male cohorts: (a) no-difference: female and male cohorts in the same population shared the same smoking rate; (b) same direction: smoking rates in female cohorts were always lower than that in male cohorts; (c) mixed direction: among the 8 sub-populations, not all female smoking rates were lower than that in male cohorts. To make these

settings more general and to cover a range of scenarios, we constructed two patterns for smoking proportions: (i) slight differences in smoking proportions between males and females but large differences in smoking proportions between populations in the same African region; (ii) large differences in smoking proportions between male and females but minor differences in smoking proportions between populations in the same African region (Supplementary Tables 4 and 5). For example, the proportions of smokers vary greatly between females and males[42], whereas other covariates may not differ as much between females and males.

To assess the performance of env-MR-MEGA at a genetic variant that is in LD with a causal variant for some populations, we simulated genotype data for 1000 pairs of genetic variants, where these genetic variants are in high LD in East Africa populations (correlation, $r = 0.85$ or $0.9$), moderate LD in West-central Africa populations ($r = 0.7$ or $0.75$), and lower LD in West Africa ($r = 0.6$ or $0.65$) and South Africa ($r = 0.45$ or $0.5$) populations. In this simulation, we considered the ancestrally homogeneous setting and replicated the second smoking pattern (ii) and same direction setting (b) (Supplementary Table 5). Under these settings, based on the simulated genotype data of 1000 causal variants and 1000 genetic variants, we compared the env-MR-MEGA tests of association and allelic heterogeneity at both the causal variant and tag SNP.

### Simulations: power evaluation
Within our simulation studies, we tested for genetic association using env-MR-MEGA, with two axes of genetic variation and sex and environmental covariates. For comparison, we also tested for association using MR-MEGA with two axes of genetic variation. The power to detect association was assessed at a genome-wide significance level ($P < 5 \times 10^{-8}$). For each environment-adjusted meta-regression model, we also obtained the power for detecting allelic heterogeneity due to ancestry and environment, as well as the power for allelic heterogeneity due to environment and the power for allelic heterogeneity due to ancestry alone. For each meta-regression model, we obtained the power to detect ancestry-correlated heterogeneity. In all replicates, power for heterogeneity was evaluated at nominal significance ($P < 0.05$).

### Application of env-adjusted MR-MEGA
To demonstrate the gains from env-MR-MEGA in assessing and controlling heterogeneity due to ancestral and environmental factors, we applied env-MR-MEGA and MR-MEGA to GWAS summary statistics for LDL-cholesterol. The GWAS data were stratified by sex and assembled from diverse sub-Saharan African cohorts, comprising a total of 19,589 participants (in AWI-Gen cohort, South African Zulu cohorts, and Uganda Genome Resource). These study cohorts have published details about the community engagement, institutional ethics review committee, and the informed consent of study participants obtained and we refer to those publications in the cohort data descriptions below.

### AWI-Gen cohort
This study includes 10,898 participants from the Africa Wits-INDEPTH partnership for Genomics studies (AWI-Gen), who passed the post-imputation filters. The AWI-Gen cohort comprises six centres in sub-Saharan African countries, including three centres in South Africa (denoted as South Africa in this study: South male = 2225, South female = 3040), one centre from Kenya (referred to as East Africa: East male = 805, East female = 961), and two centres from West Africa (one from Ghana and one from Burkina Faso, denoted as West Africa: West male = 1883, West female = 1984). EAGLE2 was used for pre-phasing, and the default Positional Burrows-Wheeler Transform algorithm was used for imputation. Imputation was performed on the cleaned dataset with 1,729,661 SNPs remaining after quality control, which included removing closely related individuals using the African Genome Resources reference panel in the Sanger Imputation Server. After imputation, poorly imputed SNPs with info scores less than 0.6, SNPs with MAF < 0.01, and HWE $p$-value < 0.00001 were excluded. The final quality-controlled imputed data had 13.98 M

SNPs. Further information on AWI-Gen cohorts has been reported in previous studies[43,44]. The study analysed the fasting serum lipids of the participants using a colorimetric assay (Randox Plus clinical chemistry analyser: Crumlin, Northern Ireland, UK). Low-density lipoprotein cholesterol (LDL-C) was then calculated using the Friedewald equation[45] and measured in mmol/L. LDL-cholesterol was regressed on age, and the first five principal components and the residuals were transformed by inverse rank normalisation. Using the residuals stratified by sex and centre, a linear mixed model association analysis was implemented using FastGWA in Genome-wide Complex Trait Analysis (GCTA version 1.91.7 beta1)[46].

### South African Zulu cohorts
The cohort of South African Zulu comprises 2707 individuals from the DDS and the Durban Diabetes Case Control Study (DCC). Previous studies have provided details on the participants' demographics, genotyping platforms, and imputation procedures[47]. After quality control, we used 16,559,897 variants and 2572 individuals. There were 1599 cases (Zulu male cases = 375; Zulu female cases = 1224) and 973 controls (Zulu male controls = 298; Zulu female controls = 675) for the association analysis. Fasting LDL-cholesterol levels (mmol/L) were measured using the ABBOTT ARCHITECT 2: CI 8200 automated analyser, which employs an enzymatic method. To conduct the association testing, we stratified the cases and controls separately by sex and regressed LDL-cholesterol levels on age, and the first two principal components; the residuals were transformed by inverse rank normalisation, and association analysis was implemented with mixed linear model-leaving one chromosome out (MLMA-LOCO) in Genome-wide Complex Trait Analysis software (GCTA version 1.91.7 beta1)[46].

### Uganda Genome resource
The Uganda Genome Resource dataset, consisting of 7000 individuals, has been documented in recent years[48,49]. This documentation covers the genotyping platforms, quality control measures, and imputation techniques. The final dataset used for genome-wide association testing, which excluded MAF < 1%, contained 15,783,409 SNPs for 6119 individuals (Uganda male = 2618; Uganda female = 3501). Non-fasting low-density lipoprotein (LDL)-cholesterol levels (mmol/L) were measured directly using the colorimetric assay described by Sugiuchi et al.[50]. LDL-cholesterol was regressed on age, and the first two principal components and the residuals were transformed by inverse rank normalisation. A sex-stratified association analysis was conducted on the residuals using mixed linear model-leaving one chromosome out (MLMA-LOCO) in Genome-wide Complex Trait Analysis (GCTA version 1.91.7 beta1)[46].

### Statistics and reproducibility
We analysed LDL cholesterol in 12 sex-stratified cohorts from East, West, and South Africa using their GWAS summary statistics. The sample sizes and environmental covariates (mean BMI, urban proportion) of each cohort are shown in Supplementary Data 1.1.

To generate axes of genetic variation, we applied a filter to exclude SNPs with MAF below 5%. Our env-adjusted MR-MEGA (env-MR-MEGA) method incorporated three variables to account for potential environmental factors that could introduce heterogeneity: sex, BMI, and participants' settings (urban or rural). As we had cohorts from different settings, we also included the proportion of participants (stratified by sex) residing in urban areas to account for potential heterogeneity in allelic effects. Our sex-stratified meta-analysis involved 12 cohorts and utilised env-MR-MEGA (for sex, BMI, sex + BMI, urban setting). The output file included $p$-values for association with LDL-cholesterol, as well as heterogeneity $p$-values due to ancestry and/or environment.

Within our analyses, we focus on forty-five genetic variants that reached genome-wide significance within the admixed African or African genetic ancestry group defined within the Global Lipids Genetic Consortium (GLGC) analyses[38] (Supplementary Data 1.2) The GLGC analysis consisted of 90,000 individuals, which is considerably larger than our analyses, so we treat their identified genome-wide significant variants in the

African-American admixed individuals as a "gold standard" and assess allelic heterogeneity at these variants that have evidence of genetic association within our cohorts.

## Reporting summary

Further information on research design is available in the Nature Portfolio Reporting Summary linked to this article.

## Data availability

The 1000 Genomes cohorts used in this paper are available from the developers of MAGMA[51] (https://vu.data.surfsara.nl/index.php/s/ePXET6IWVTwTes4/download). All individual level data (phenotype, genotype) are available to researchers under managed access from the EGA database under the study accession codes EGAS00001001558, EGAD00010000965, EGAS00001002482, EGAD00001006425, and EGAD00010001996. The availability of the AWI-Gen datasets is subject to controlled access through, the Data and Biospecimen Access Committee of the H3Africa Consortium. GLGC summary statistics for lead SNPs of the lipid traits are available from Supplementary Table 3 of Graham et al. [38] [https://doi.org/10.1038/s41586-021-04064-3]. The summary statistics from our env-MR-MEGA analyses of twelve sex-stratified GWAS from Africa, at the genome-wide significant variants for LDL in the African-American meta-analysis from GLGC, are available in Supplementary Data 1.2.

## Code availability

Our env-MR-MEGA method, env-MR-MEGA[52], is freely available as an R library at https://github.com/SiruRooney/env.MRmega.

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

## Acknowledgements

This research is funded by the UK Medical Research Council (MR/W02098X/1). SW and JA are also funded by the UK Medical Research Council (MC_UU_00002/4). For the purpose of Open Access, the authors have applied a CC BY public copyright licence to any Author Accepted Manuscript version arising from this submission.

## Author contributions

J.L.A., T.C., S.F., and A.P.M. conceived and designed the project. J.L.A. co-ordinated the project. S.W. developed the env-MR-MEGA method and software with input from J.L.A. and A.P.M. S.W. and O.O.O. performed statistical analyses with input from J.L.A., A.P.M., T.C., S.F., and M.R. T.C., A.K., and O.O.O. performed the sex-stratified GWAS. M.R. and S.F. contributed data. S.W. and O.O.O. wrote the first draft of the manuscript with input from J.L.A., T.C., S.F., A.P.M., M.R., and A.K. All authors approved the final version of the manuscript.

## Competing interests

The authors declare no competing interests.
