## [Transparent Peer Review file · Communications Biology]

Accounting for heterogeneity due to environmental sources in meta-analysis of genome-wide association studies

Corresponding Author: Dr Jennifer Asimit

Version 0:

Reviewer comments:

Reviewer #1

(Remarks to the Author)

This manuscript by Wang et al present an environment-adjusted meta-regression method (env-MR-MEGA) for identifying genetic associations using summary statistics using GWAS summary statistics. The method is a natural extension of the MR-MEGA and simulations and real data analysis suggest Env-MR-MEGA can be more powerful than MR-MEGA in some cases. Env-MR-MEGA can be very helpful in interpreting genetic associations of variants. The manuscript is well written, and I only have comments for improving the manuscript.

- 1) The model simulated in simulation study 1 can also be viewed as gene-sex interaction. However, this paper considers it as heterogeneity in allelic effect. Could authors explain whether heterogeneity studied in this study is equivalent to gene-environment interactions? Any clarification will be helpful.
- 2) In the simulation study 2 (page 16), what are the smoking specific allelic effects? I could not see the phenotype simulation model such as listed in Supplementary Material Table S1.
- 3) When a genetic variant is in LD with an unknown causal variant and the LD pattern and allele frequencies vary across ancestries, I would expect heterogeneity occurs even the effect size of the causal variant does not change across populations. In this case, Env-MR-MEGA may identify the association. I am wondering if authors still interpret it as heterogeneity.
- 4) Relevant to my comment (3). It seems the simulations only consider the causal variants. It would be nice to see the performance of Env-MR-MEGA when a genetic variant is in LD with a causal variant and the LD pattern and frequencies of testing variant and causal variant across populations.
- 5) In the LDL-C analysis result table 1, the authors applied a threshold $P < 0.05$ for significance. Should multiple tests be addressed?
- 6) A recent study using GLGC and gene-lifestyle interaction summary statistics to search for gene-environment interactions (PMID: 38649715). They reported significant gene-smoking or gene-alcohol drinking interactions for LDL-C near the loci rs12740374, rs1712248, rs138294113 and rs7412 reported in this study. Although the current study does not directly address gene-environment interactions, the heterogeneity could be partially explained by the interactions and some discussion would be interesting.

Reviewer #2

(Remarks to the Author)

Comments to the Authors

The authors proposed a new method to account for environmental effects using GWAS summary statistics based on the frame work of MR.MEGA. This paper is easy to follow in general and presented interesting results to show power gained in signals identification across diverse populations in simulation studies. To improve this manuscript, I have some comments as below:

Major comments:

1. Line 162-163: It is not clear how type I errors were calibrated at 0.05 from Table S4.
2. For the real data analysis, it is not clear how GWAS were carried out in individual cohort to generate summary statistics as input for env-MR-MEGA. In the supplementary Data1, only results from env-MR-MEGA were presented but without any comparison with results by other methods. I think it would be good to see some more work to show the credibility for the results by env-MR-MEGA. Would effect sizes estimated from env-MR-MEGA improve the PRS prediction in African populations?

Minor comments:

1. Supplementary Tables/Figures were not referenced in correct order.

Reviewer #3

(Remarks to the Author)

Preamble

Meta-analysis of genome-wide association studies (GWAS) across different populations would improve power in finding disease-associated loci without access to individual-level data. However, when there exists effect size heterogeneity among different genetic ancestries and environments, the metaanalysis with fixed-effect models cannot accommodate such heterogeneity while classic random-effect models have lower power under structural heterogeneity with respect to ancestry and environment. To take both environmental and ancestral heterogeneity into account, an environment-adjusted metaregression (env-MR-MEGA) model is proposed to detect both genetic associations and allelic heterogeneity. Simulation of multiple scenarios show non-inferior power of the env-MR-MEGA model over the model not adjusting for environment in both detecting genetic association and allelic heterogeneity. The power gain is great especially when the heterogeneity is strongly correlated with environmental factors. Application of the env-MR-MEGA to twelve sex-stratified African GWAS detects 9 SNPs with effect size heterogeneity with respect to environmental factors.

In general, this is an interesting paper working on an important topic. However, there are several issues needed to be addressed. Major Comments

1. In the Section "Materials and Methods", the chi-square test statistics are introduced for multiple nulls for a variant, including

- null of no associations across all cohorts (degree of freedom $T + S + 1$);
- null of absence of ancestral and environmental heterogeneity (degree of freedom $T + S$);
- null of absence of ancestral heterogeneity (degree of freedom T);
- null of absence of environmental heterogeneity (degree of freedom S);
- null of absence of residual heterogeneity (degree of freedom $K - S - T - 1$).

However, the proposed env-MR-MEGA model is a one-shot approach that can only infer one of the first four nulls. If the null of no associations has been rejected, the type-I error rate would inflate if we directly test nulls of absence of heterogeneity on the same data, which is a natural subsequent task of mega analysis. I believe that such a declaration is a necessity and it would be much better to include some intuitive discussions on the sequential inference of genetic associations and allelic heterogeneity.

2. In Supplementary Table S5 which present the empirical type-I error rate of the proposed envMR-MEGAmoel and the MR-MEGA model in detecting allelic heterogeneity, it is weird that the empirical type-I error rate of both models remains the same across different association 1levels (β) without sampling variation. Do you perform analysis on different 1000 random datasets under different association levels (β)? If so, I think it is impossible to have exactly the same empirical type-I error rate. If not, please add the declaration on how you perform simulation in evaluating the type-I error rate of allelic heterogeneity detection.

3. In lines 292 to 296, you make a declaration with respect to Fig S6-S8 that "Especially for the South Africa scenario, where the heterogeneity in allelic effect is more strongly correlated with the South Africa ancestry compared to smoking exposures, the power for allelic heterogeneity due to ancestry attained from both env-MR-MEGA and MR-MEGA is noticeably greater than that due to ancestry and environment." However, the difference between power of env-MRMEGA-AHet and MR-MEGA-AHet and power of env-MR-MEGA-AEHet in Fig S6-S8 is not that noticeable as declared. Can you add more explanation, such as adding error bar for in the Figures to visualize the significant difference in power?

4. In lines 305 to 309, you make a declaration with respect to Fig S9-S10 that "In addition, compared to the mixed direction setting where some female cohorts have higher smoking proportions than male cohorts in the same populations, the same direction in smoking proportions gained more power for heterogeneity in environmental effects than mixed direction." However, this is not true for the East Africa scenario, where the power of env-MR-MEGA-EHet is higher under the mixed direction setting. In addition, according to your Table S3, the West Africa scenario is the same between the same direction setting and the mixed direction setting. I think this makes it weird to have different simulation results for the West Africa scenario between Fig S9-S10.

5. In lines 359 to 360, you make a declaration with respect to Table 1 that "Three SNPs displayed heterogeneity due to ancestry alone (rs11576986,rs1712248 and rs138294113)". However, I don't know why you conclude that rs1712248 displayed heterogeneity due to ancestry alone, given that the p-value of its heterogeneity due to both ancestry and an environmental effect is smaller in most of cases.

Minor Comments

1. The tick mark labels for all Figures are too small and hard to read as a result; they should be larger. In addition, can you adjust colors of points in Figures 4(a) and S1 such that points of the same clusters are of similar colors for better interpretation?
2. Although p-values that are smaller than the target significance level (0.05) are highlight in bold in Table 1, it is still hard to identify significant p-values at the first glimpse. Can you add asterisk as superscript to those significant p-values for better presentation?
3. This paper warrants further polishing in writing and more attentions should be paid on grammar issues.

Decision With the above concerns, I would recommend a "Reject & Resubmit" decision on this paper.

Version 1:

Reviewer comments:

Reviewer #1

(Remarks to the Author)

I am satisfied with the revision and have no further questions.

Reviewer #2

(Remarks to the Author)

Thanks for the improved version of manuscript. I am happy with the response to my comments.

Referee expertise:

Referee #1: Statistical genetics, methods development

Referee #2: Statistical genetics, integration of public health data, GWAS

Referee #3: Statistical genetics, integration of public health data, GWAS

We greatly appreciate the valuable and insightful feedback provided by the reviewers. Their constructive comments have been instrumental in enhancing the quality of our manuscript.

Below, we have outlined our responses to the reviewers' remarks.

Reviewers' comments:

Reviewer #1 (Remarks to the Author):

This manuscript by Wang et al present an environment-adjusted meta-regression method (env-MR-MEGA) for identifying genetic associations using summary statistics using GWAS summary statistics. The method is a natural extension of the MR-MEGA and simulations and real data analysis suggest Env-MR-MEGA can be more powerful than MR-MEGA in some cases. Env-MR-MEGA can be very helpful in interpreting genetic associations of variants. The manuscript is well written, and I only have comments for improving the manuscript.

1) The model simulated in simulation study 1 can also be viewed as gene-sex interaction. However, this paper considers it as heterogeneity in allelic effect. Could authors explain whether heterogeneity studied in this study is equivalent to gene-environment interactions? Any clarification will be helpful.

Response: We thank the reviewer for this insightful question. Simulating from a gene-sex interaction model introduces allelic heterogeneity that we assess with env-MR-MEGA, though we do not directly assess gene-environment interactions. We have clarified this in the second paragraph of the Results section of the general model framework:

“We have designed env-MR-MEGA with two aims: (i) adjusting for environmental exposures and ancestral effects when testing if a variant is associated with a trait across GWAS; (ii) assessing whether there are environmental and/or ancestral effects that impact the association and are source(s) of allelic heterogeneity. When there is allelic heterogeneity due to environment, this suggests presence of gene-environment interactions, though this interaction is not directly assessed by env-MR-MEGA since we

do not estimate/test the interaction effect. Our modelling framework allows us to assess the impact of environmental exposures on allelic heterogeneity using only study-level summaries of environmental variables, without the need for individual-level data.”

2) In the simulation study 2 (page 16), what are the smoking specific allelic effects? I could not see the phenotype simulation model such as listed in Supplementary Material Table S1.

Response: Thank you for pointing this out and we have added clarification. In simulation 1, we considered the model of female-specific association: $Y = \varepsilon$ for males and $Y = \beta \times a + \varepsilon$ for females, where β is effect size for the genetic variant. In simulation 2, we considered the model of smoker-specific association: $Y = e \times (\beta \times g) + \varepsilon$; where β is allelic effect of the causal variant and $e = 1$ for smokers and $e = 0$ for non-smokers – this same model is used for both sexes, but we consider different settings for smoker proportions for each sex in each population. Simulations 1 and 2 both use the same allelic effect patterns that are provided in Table S1.

We have clarified the caption of Table S1, adding:

“These allelic effect patterns are used in simulation 1 for gene-sex interactions and simulation 2 for gene-smoker interactions, where both genders have the same model but differ in smoking proportions within each population.”

And we have added the following to the Results section, in the Simulation 2 subsection:

“In this simulation, we use smoker-specific allelic effect patterns for the six heterogeneity scenarios previously described for sex-specific allelic effect simulations (Supplementary Material Table S1); the same pattern is used for males and females and the effect size varies from 0 to 0.2 with increments of 0.02.”

3) When a genetic variant is in LD with an unknown causal variant and the LD pattern and allele frequencies vary across ancestries, I would expect heterogeneity occurs even the effect size of the causal variant does not change across populations. In this case, Env-MR-MEGA may identify the association. I am wondering if authors still interpret it as heterogeneity.

Response: Thank you for another interesting question. We agree that this would be interpreted as heterogeneity. Since we test heterogeneity at each variant, in this situation, we expect that heterogeneity would be detected at the non-causal variant that is a tag SNP of the causal variant (also see response to the next question). We have added the following to paragraph 4 of Results section, on page 4:

“Tag SNPs, being in LD with the causal variant, are likely to also be detected as associated with the trait by env-MR-MEGA. In the ancestrally homogeneous setting,

where the causal variant has the same effect size in all populations, it is plausible that a tag SNP will have different levels of correlation with the causal variant due to different population LD patterns. In that case, env-MR-MEGA is expected to detect allelic heterogeneity due to ancestry at the tag SNP, but not the causal variant – as the causal variant is unknown in practice, we would consider the tag SNP to have allelic heterogeneity. We explore this in a simulation scenario where there is ancestral homogeneity and varying environmental exposure proportions.”

4) Relevant to my comment (3). It seems the simulations only consider the causal variants. It would be nice to see the performance of Env-MR-MEGA when a genetic variant is in LD with a causal variant and the LD pattern and frequencies of testing variant and causal variant across populations.

Response: Thank you for the valuable suggestion and we have added in a simulation where both a causal variant and a variant in LD with it are tested by env-MR-MEGA. Using a bivariate binomial distribution, we simulated the genotype data of a genetic variant which has different levels of LD with the causal variant amongst the populations. Then under the setting where female smoking proportions were significantly lower than male smoking proportions in the ancestrally homogeneous scenario, the traits were generated by the model of smoker-specific association. More details of the new simulation design are shown in the Materials and Methods section, Simulation 2 subsection, second paragraph on page 18.

*“To assess the performance of env-MR-MEGA at a genetic variant that is in LD with a causal variant for some populations, we simulated genotype data for 1000 pairs of genetic variants, where these genetic variants are in high LD in East Africa populations (correlation, $r=0.85$ or 0.9), moderate LD in West-central Africa populations ($r=0.7$ or 0.75), and lower LD in West Africa ($r=0.6$ or 0.65) and South Africa ($r=0.45$ or 0.5) populations. In this simulation, we considered the ancestrally homogeneous setting and replicated the second smoking pattern (ii) and same direction setting (b) (**Supplementary Material Table S5**). Under these settings, based on the simulated genotype data of 1000 causal variants and 1000 genetic variants, we compared the env-MR-MEGA tests of association and allelic heterogeneity at both the causal variant and tag SNP.”*

The results of the new simulation where a genetic variant in LD with a causal variant in some populations is involved in environment-adjusted meta-analysis are shown before Fig 2.

“Additionally, we investigated the performance of env-MR-MEGA at a tag SNP of a causal variant, where the LD between the causal variant and tag SNP is similar in populations from the same region and varies between regions (Methods). In the ancestrally homogeneous scenario, we replicated the smoking proportions across 16 sex-stratified

cohorts in the second smoking pattern (b) in the same direction setting (Supplementary Material Table S5). Sequentially, we generated 1000 replicates of genotype data for a causal variant and its correlated genetic variant. We then applied env-MR-MEGA to each of the 1000 causal variants and 1000 tag SNPs to estimate the power for association and heterogeneity due to ancestry and/or environment at causal variants and at tag SNPs.

It is demonstrated that, when the allelic effect varied between 0.02 and 0.1, the power for association at causal variants is notably greater than that from genetic variants that are in LD with causal variants (different LD amongst populations) (**Supplementary Material Fig S12**). Likewise at low-moderate effect sizes of 0.02 to 0.1, power to detect allelic heterogeneity due to ancestry and environment or due to only environment, is notably higher at the causal variants compared to the tag SNPs. As expected, the power for heterogeneity due to ancestry attained the nominal significance level for the causal variants. However, at the tag SNPs, the power for ancestral heterogeneity increased with the size of the allelic effect. This is due to the lower correlation with the causal variant in populations from West Africa and South Africa.”

5) In the LDL-C analysis result table 1, the authors applied a threshold $P < 0.05$ for significance. Should multiple tests be addressed?

Response: This is a good point, and we have adjusted our analysis based on this suggestion and the first comment from Reviewer 3. We only considered 45 variants that are genome-wide significant in the African ancestries from the GLGC study and present in our data. Amongst these 45 variants, we adjust for multiple testing to detect association ($p < .05/45 = .001$) and then only test for heterogeneity in those variants with evidence of association at 0.001 and a suggestive threshold of 0.005. This resulted in the removal of three variants from Table 1, which did not have evidence of genetic associations in our data, but had evidence of heterogeneity (one due to ancestry only and two from sex and urban with/without ancestry).

We have inserted the following in section identification of sources of genetic heterogeneity in Africa cohorts:

“We considered 45 SNPs that have previously attained genome-wide significance for LDL-cholesterol in the meta-analysis of the substantially larger African American GWAS from GLGC³⁸ and are present in our data. For these SNPs, we used env-MR-MEGA and MR-MEGA to first test for association in our African cohorts using a Bonferroni-corrected threshold for association of 0.001 (.05/45) and a suggestive threshold of 0.005. Amongst the SNPs that env-MR-MEGA/MR-MEGA identified as associated in our data, we investigated the contribution of ancestry and environmental variables (sex, BMI, and urban environment) to heterogeneity in allelic effects using a nominal threshold of 0.05 (as detection of association has been adjusted for multiple tests).”

And updated our results, including Table 1, accordingly:

“Ancestral heterogeneity is detected by both MR-MEGA and env-MR-MEGA at five variants (rs12740374, rs1712248, rs11884143, rs4245791, rs7412), and env-MR-MEGA detects ancestral heterogeneity at an additional two variants (rs138294113, rs3808607) (Supplementary Data 1). In addition, env-MR-MEGA shows evidence of heterogeneity due to ancestry and environment at six of these variants; at rs138294113 env-MR-MEGA only detects ancestral heterogeneity. Our env-MR-MEGA method detected allelic heterogeneity involving environmental effects in seven SNPs: one SNP driven by sex (rs12740374), one SNP jointly driven by combined sex and BMI (rs4245791), and five SNPs driven by urban status (rs28362286, rs11884143, rs3735018, rs3808607, and rs7412).”

6) A recent study using GLGC and gene-lifestyle interaction summary statistics to search for gene-environment interactions (PMID: 38649715). They reported significant gene-smoking or gene-alcohol drinking interactions for LDL-C near the loci rs12740374, rs1712248, rs138294113 and rs7412 reported in this study. Although the current study does not directly address gene-environment interactions, the heterogeneity could be partially explained by the interactions and some discussion would be interesting.

Response: Thank you for your comment. We have explained how our method differs from the recently published gene-environment interaction method. We have also discussed the overlap in the loci detected in these two methods. Below is a new paragraph inserted in the fifth paragraph of the Discussion section:

“Complex traits are influenced by a combination of genetic and environmental factors; thus, genome-wide interaction studies (GWIS) have been modelled to understand how genetic variants modify the effect of environmental exposures on a disease or trait (Laville et al., 2022; Zhu et al., 2024). Our method, env-MR-MEGA, differs from GWIS, specifically targeting environmental heterogeneity to reduce confounding and identify genetic association signals. In a recent investigation into gene-environment interactions involving LDL-cholesterol, independent loci near CELSR2: rs12740374, APOB: rs1712248, APOE: rs7412 and LDLR/SMARCA4: rs138294113 which we identified in our application to GLGC, have been shown to interact with two environmental factors (cigarette smoking and alcohol consumption) and contributed to heterogeneity in allelic effects across different populations (Zhu et al., 2024). Thus, underscoring the consistency and environmental influences on these genetic association signals across different methodological approaches.”

Reviewer #2 (Remarks to the Author):

Comments to the Authors

The authors proposed a new method to account for environmental effects using GWAS summary statistics based on the framework of MR.MEGA. This paper is easy to follow in general and presented interesting results to show power gained in signals identification across diverse populations in simulation studies. To improve this manuscript, I have some comments as below:

Major comments:

1. Line 162-163: It is not clear how type I errors were calibrated at 0.05 from Table S4.

Response: Thanks for pointing out this typo. The type I errors were estimated using a nominal threshold of 0.05, whereas a genome-wide threshold was used for power. We have updated the caption of the table (re-labelled as S2):

“False positive error rates (Type I errors), at a nominal significance threshold ($P < 0.05$), to detect association from env-MR-MEGA and MR-MEGA across a range of heterogeneity scenarios are well-calibrated. The type I errors were estimated as the probability that the causal variant has P-value of association less than 0.05, based on 1000 replications.”

2. For the real data analysis, it is not clear how GWAS were carried out in individual cohort to generate summary statistics as input for env-MR-MEGA. In the supplementary Data1, only results from env-MR-MEGA were presented but without any comparison with results by other methods. I think it would be good to see some more work to show the credibility for the results by env-MR-MEGA.

Would effect sizes estimated from env-MR-MEGA improve the PRS prediction in African populations?

Response: Thank you for this comment and we have provided more details in the cohort descriptions within the Materials and Methods section. In summary, we used mixed linear model methods implemented in GCTA for all the cohorts with FastGWA for AWI-Gen cohorts and mixed linear model-leaving one chromosome out (MLMA-LOCO) for the Zulu and Uganda genome resource cohorts.

In Supplementary Data 1, we have added in results from applying MR-MEGA to our data, and discuss these results in the Results section:

“Ancestral heterogeneity is detected by both MR-MEGA and env-MR-MEGA at five variants (rs12740374, rs1712248, rs11884143, rs4245791, rs7412) and env-MR-MEGA detects ancestral heterogeneity at an additional two variants (rs138294113, rs3808607) (Supplementary Data 1).”

With regards to PRS, we have add this to the Discussion section:

“We expect that env-MR-MEGA will improve PRS prediction in African populations, which we plan to implement in future work”

Minor comments:

1. Supplementary Tables/Figures were not referenced in correct order.

Response: Thank you for pointing this out. We have re-ordered the tables and figures of Supplementary material so that they are numbered in order of appearance in the manuscript.

Reviewer #3 (Remarks to the Author):

Preamble

Meta-analysis of genome-wide association studies (GWAS) across different populations would improve power in finding disease-associated loci without access to individual-level data. However, when there exists effect size heterogeneity among different genetic ancestries and environments, the metaanalysis with fixed-effect models cannot accommodate such heterogeneity while classic random-effect models have lower power under structural heterogeneity with respect to ancestry and environment. To take both environmental and ancestral heterogeneity into account, an environment-adjusted metaregression (env-MR-MEGA) model is proposed to detect both genetic associations and allelic heterogeneity. Simulation of multiple scenarios show non-inferior power of the env-MR-MEGA model over the model not adjusting for environment in both detecting genetic association and allelic heterogeneity. The power gain is great especially when the heterogeneity is strongly correlated with environmental factors. Application of the env-MR-MEGA to twelve sex-stratified African GWAS detects 9 SNPs with effect size heterogeneity with respect to environmental factors.

In general, this is an interesting paper working on an important topic. However, there are several issues needed to be addressed.

Major Comments

1. In the Section “Materials and Methods”, the chi-square test statistics are introduced for multiple nulls for a variant, including

- null of no associations across all cohorts (degree of freedom $T + S + 1$);
- null of absence of ancestral and environmental heterogeneity (degree of freedom $T + S$);
- null of absence of ancestral heterogeneity (degree of freedom T);

- null of absence of environmental heterogeneity (degree of freedom S);
- null of absence of residual heterogeneity (degree of freedom $K - S - T - 1$).

However, the proposed env-MR-MEGA model is a one-shot approach that can only infer one of the first four nulls. If the null of no associations has been rejected, the type-I error rate would inflate if we directly test nulls of absence of heterogeneity on the same data, which is a natural subsequent task of mega analysis. I believe that such a declaration is a necessity and it would be much better to include some intuitive discussions on the sequential inference of genetic associations and allelic heterogeneity.

Response: Thank you for this insightful comment. Considering this comment and comment 5 of Reviewer 1, we have adjusted our real data analysis strategy. As in our initial analysis, we focus on 45 variants that are genome-wide significant in the substantially larger GLGC African ancestry analysis. However, before assessing evidence for sources of heterogeneity at these variants, we first test the variants for association and only assess heterogeneity at variants with evidence of association. This resulted in the removal of three variants from Table 1, which did not have evidence of genetic associations in our data, but appeared to have evidence of heterogeneity (one due to ancestry only and two from sex and urban with/without ancestry).

Considering our simulation results that estimate type 1 error for rejecting the null of heterogeneity from different sources (Table S3), we believe that our tests are well-calibrated when used only at variants with evidence of association.

We have added the following to the beginning of the application section in the Results section (page 11):

“We considered 45 SNPs that have previously attained genome-wide significance for LDL-cholesterol in the meta-analysis of the substantially larger African American GWAS from GLGC³⁸ and are present in our data. For these SNPs, we used env-MR-MEGA and MR-MEGA to first test for association in our African cohorts using a Bonferroni-corrected threshold for association of 0.001 (.05/45) and a suggestive threshold of 0.005. Amongst the SNPs that env-MR-MEGA/MR-MEGA identified as associated in our data, we investigated the contribution of ancestry and environmental variables (sex, BMI, and urban environment) to heterogeneity in allelic effects using a nominal threshold of 0.05 (as detection of association has been adjusted for multiple tests).

We note that testing for heterogeneity at variants that have evidence of association does not inflate the type 1 error of these heterogeneity tests. This is because we include the ancestral and environmental covariates in our model to test for association, without any previous testing for significance of these covariates. If we had first tested the sources of heterogeneity and then only included in our model the sources that have evidence of heterogeneity, then we encounter the problem of forking paths and

consequently, inflated type 1 error. In the forking path problem, variables are first tested and then included in the model if they are found to have a significant effect, assuming that otherwise that variable would not have been included in the model (Rubin, 2023)."

2. In Supplementary Table S5 which present the empirical type-I error rate of the proposed envMR-MEGAmoel and the MR-MEGA model in detecting allelic heterogeneity, it is weird that the empirical type-I error rate of both models remains the same across different association levels (β) without sampling variation. Do you perform analysis on different 1000 random datasets under different association levels (β)? If so, I think it is impossible to have exactly the same empirical type-I error rate. If not, please add the declaration on how you perform simulation in evaluating the type-I error rate of allelic heterogeneity detection.

Response: Thank you for pointing out this observation and we have now added more detail to clarify how we obtained our results.

In the simulation design, we randomly selected 1000 causal variants and simulated the genotype data under the Hardy-Weinberg equilibrium (HWE) assumption in all populations using the population-specific allele frequency for the population. We simulated quantitative traits from the model $Y = \beta \times g + \varepsilon$, where g is the simulated genotype at a variant and we vary the allelic effect β . For each value of the allelic effect β , the same 1000 causal variants and their simulated values were used to generate new Y values in each setting. Within each setting of the effect size β we set a sequence of seeds for generation of the error term in our simulation of Y . This same sequence of seeds was used for each setting of β . Therefore, the estimated allelic effect in the simulated GWAS file would increase at the same magnitude when the allelic effect β increased by 0.02. This results in the same empirical type I errors under different allelic effect sizes. We have added a detailed description of our simulation design in the Methods section on Simulation 1, the third paragraph of page 17:

"To evaluate the power and type I errors for each heterogeneity scenario, in a series of simulations, we first selected 1000 causal variants with $MAF > 1\%$. Then, simulated the genotype data of the 1000 causal variants under the Hardy-Weinberg equilibrium (HWE) assumption in all populations using the population-specific allele frequency. Under each allelic effect β , the quantitative traits were simulated from the same 1000 causal variants and their generated genotype data. In addition, within each setting of the effect size β we set a sequence of seeds for generation of the error term in our simulation of Y . This same sequence of seeds was used for each setting of β . As a baseline power comparison between the two methods, we considered equal-sized cohorts (1000 in each female/male cohort) and then unequal sample sizes."

3. In lines 292 to 296, you make a declaration with respect to Fig S6-S8 that “Especially for the South Africa scenario, where the heterogeneity in allelic effect is more strongly correlated with the South Africa ancestry compared to smoking exposures, the power for allelic heterogeneity due to ancestry attained from both env-MR-MEGA and MR-MEGA is noticeably greater than that due to ancestry and environment.” However, the difference between power of env-MRMEGA-AHet and MR-MEGA-AHet and power of env-MR-MEGA-AEHet in Fig S6-S8 is not that noticeable as declared. Can you add more explanation, such as adding error bar for in the Figures to visualize the significant difference in power?

Response: Thanks for offering this valuable suggestion. To make the gains in power for allelic heterogeneity due to ancestry over that due to ancestry and environment clear, we focus on the South Africa scenario and make a plot summarising the power for allelic heterogeneity across the three settings of smoking proportions (no difference, same direction and mixed direction) and add error bars for each power estimate. The added plot is in Supplementary Material and labeled as Figure S9. From Fig S9, it shows that, for the South Africa scenario, the power for allelic heterogeneity due to ancestry attained from both env-MR-MEGA and MR-MEGA is moderately greater than that due to ancestry and environment. A more detailed description for Fig S9 is shown in the first paragraph of page 8.

“For the South Africa scenario, where the heterogeneity in allelic effect is more strongly correlated with South Africa ancestry compared to smoking exposures, the power for allelic heterogeneity due to ancestry attained from both env-MR-MEGA and MR-MEGA is moderately greater than that due to ancestry and environment. For the no difference setting and mixed direction setting in the South Africa scenario, where the differences in smoking proportions between male and females are not consistent with sex effects, the power for allelic heterogeneity due to ancestry is obviously greater than that due to ancestry and environment when the allelic effect of the causal variant varied between 0.04 and 0.06. For the same direction setting, where female cohorts have lower smoking proportions, detection of allelic heterogeneity due to ancestry and environment had clear gains in power over ancestry alone (MR-MEGA and env-MR-MEGA) when there is a moderate allelic effect, around 0.06 (Supplementary Material Fig S9).”

4. In lines 305 to 309, you make a declaration with respect to Fig S10-S11 (previous Fig S9-S10) that “In addition, compared to the mixed direction setting where some female cohorts have higher smoking proportions than male cohorts in the same populations, the same direction in smoking proportions gained more power for heterogeneity in environmental effects than mixed direction.” However, this is not true for the East Africa scenario, where the power of env-MR-MEGA-EHet is higher under the mixed direction setting. In addition, according to your Table S5 (previous Table S3), the West Africa scenario is the same between the same direction setting and the mixed direction setting.

I think this makes it weird to have different simulation results for the West Africa scenario between Fig S10-S11 (previous Fig S9-S10).

Response: Thank you for these helpful comments and observations. We have added more detail and clarifications to the Results section on Page 8:

“In the mixed direction setting, where for the South Africa and West-central Africa regions, one of the two populations had a higher smoking proportion for females than males (Supplementary Material Table S5), there is a loss in power to detect environmental heterogeneity over the setting where both populations in a region have the same smoking pattern between sexes (Figures S10, S11). Also, we note that in the mixed direction setting, we randomly selected populations to have a higher proportion of female smokers than male. This resulted in another version of the “same” setting for the East Africa region, where both populations have a higher proportion of female smokers. Consequently, the East Africa region has a higher power to detect environmental heterogeneity in the mixed simulations than in the same setting simulations, where female smoker proportions were lower in both populations. Finally, in the mixed direction setting, the West Africa female smoker proportions remain identical to the same direction setting. However, there are slight differences between the power estimates between these two settings for West Africa. Although the data generated under the same and mixed settings use the same smoking proportions for West Africa and there is only a smoking effect for West Africa populations, the observed difference arises from our application of env-MR-MEGA. In applying env-MR-MEGA, we use the smoking proportions from all populations under the same or mixed setting, for which there are several slight differences due to setting the females to have a higher smoker proportion in some populations. Consequently, the power to detect environmental heterogeneity is similar in the same and mixed settings. In contrast, the power to detect ancestral heterogeneity is slightly lower in the same direction setting compared to the mixed setting. This is explained by the smaller chi-squared statistic that results from similar female smoking proportions in the same direction setting, compared to the larger differences in the mixed setting.”

5. In lines 359 to 360, you make a declaration with respect to Table 1 that “Three SNPs displayed heterogeneity due to ancestry alone (rs11576986,rs1712248 and rs138294113)”. However, I don’t know why you conclude that rs1712248 displayed heterogeneity due to ancestry alone, given that the p-value of its heterogeneity due to both ancestry and an environmental effect is smaller in most of cases.

Response: Thank you for your observation and we have taken this into account in a more detailed description that also accounts for comment 2 of Reviewer 2. The following has been added to the Results section, prior to Fig 4.

“Ancestral heterogeneity is detected by both MR-MEGA and env-MR-MEGA at five variants (rs12740374, rs1712248, rs11884143, rs4245791, rs7412), and env-MR-MEGA detects ancestral heterogeneity at an additional two variants (rs138294113, rs3808607)

(Supplementary Data 1). In addition, env-MR-MEGA shows evidence of heterogeneity due to ancestry and environment at six of these variants; at rs138294113 env-MR-MEGA only detects ancestral heterogeneity.”

Minor Comments

1. The tick mark labels for all Figures are too small and hard to read as a result; they should be larger. In addition, can you adjust colors of points in Figures 4(a) and S1 such that points of the same clusters are of similar colors for better interpretation?

Response: Thank you for this clarifying comment that improves the figures. We have adjusted all figures in the main paper and supplementary material, accordingly.

2. Although p-values that are smaller than the target significance level (0.05) are highlight in bold in Table 1, it is still hard to identify significant p-values at the first glimpse. Can you add asterisk as superscript to those significant p-values for better presentation?

Response: As suggested, we have included asterisks as a superscript in addition to the bold font. The following has been added to the caption of Table 1:

“At GLGC variants that meet the Bonferroni-adjusted threshold (0.001) of genetic association in our data, we highlight heterogeneity p-values < 0.05 by presenting them with two asterisks in bold font. At GLGC variants that meet the suggestive threshold (0.005) of genetic association, we highlight heterogeneity p-values < 0.05 by presenting them with one asterisk in bold font.”

3. This paper warrants further polishing in writing and more attentions should be paid on grammar issues.

We have carefully revised our paper to address this. All changes have been tracked.

REVIEWERS' COMMENTS:

Reviewer #1 (Remarks to the Author):

I am satisfied with the revision and have no further questions.

Reviewer #2 (Remarks to the Author):

Thanks for the improved version of manuscript. I am happy with the response to my comments.

RESPONSE: We thank the reviewers for their constructive feedback, which has improved our manuscript.